# Cervical transcutaneous vagal neuromodulation in chronic pancreatitis patients with chronic pain: A randomised sham controlled clinical trial

**Janusiya Anajan Muthulingam**[1,2], **Søren Schou Olesen**[2,3], **Tine Maria Hansen**[1,2], **Christina Brock**[2,4], **Asbjørn Mohr Drewes**[2,3], **Jens Brøndum Frøkjær**[1,2]*

**1** Mech-Sense, Department of Radiology, Aalborg University Hospital, Aalborg, Denmark, **2** Department of Clinical Medicine, Aalborg University, Aalborg, Denmark, **3** Department of Gastroenterology & Hepatology, Centre for Pancreatic Diseases, Aalborg University Hospital, Aalborg, Denmark, **4** Mech-Sense, Department of Gastroenterology and Hepatology, Aalborg University Hospital, Aalborg, Denmark

* jebf@rn.dk

**Data Availability Statement:** Data is available at https://doi.org/10.6084/m9.figshare.14009030.

**Funding:** This study was supported by Independent Research Fund Denmark (DFF –

## Abstract

### Background & aims

Chronic abdominal pain is the primary symptom of chronic pancreatitis, but unfortunately it is difficult to treat. Vagal nerve stimulation studies have provided evidence of anti-nociceptive effect in several chronic pain conditions. We investigated the pain-relieving effects of transcutaneous vagal nerve stimulation in comparison to sham treatment in chronic pancreatitis patients.

### Methods

We conducted a randomised double-blinded, sham-controlled, crossover trial in patients with chronic pancreatitis. Patients were randomly assigned to receive a two-week period of cervical transcutaneous vagal nerve stimulation using the gammaCore device followed by a two-week sham stimulation, or vice versa. We measured clinical and experimental endpoints before and after each treatment. The primary clinical endpoint was pain relief, documented in a pain diary using a visual analogue scale. Secondary clinical endpoints included Patients' Global Impression of Change score, quality of life and Brief Pain Inventory questionnaire. Secondary experimental endpoints included cardiac vagal tone and heart rate.

### Results

No differences in pain scores were seen in response to two weeks transcutaneous vagal nerve stimulation as compared to sham treatment (difference in average pain score (visual analogue scale): 0.17, 95%CI (-0.86;1.20), P = 0.7). Similarly, no differences were seen for secondary clinical endpoints, except from an increase in the appetite loss score (13.9, 95% CI (0.5;27.3), P = 0.04). However, improvements in maximum pain scores were seen for transcutaneous vagal nerve stimulation and sham treatments as compared to their

7016-00073), Axel-Muusfeldts Foundation, and Læge Sofus Carl Emil Friis og Hustru Olga Doris Friis' Legat. Jens Brøndum Frøkjær received all the grants. The funders had no role in study design, data collection and analysis, decision to publish, or preparation of the manuscript.

**Competing interests:** The authors have declared that no competing interests exist.

**Abbreviations:** BPI, Brief pain inventory; CP, Chronic pancreatitis; CVT, Cardiac vagal tone; LVS, Linear vagal scale; PGIC, Patients' Global Impression of Change score; nVNS, non-invasive vagal nerve stimulation; VAS, Visual analogue scale; VNS, Vagal nerve stimulation.

respective baselines: vagal nerve stimulation (-1.3±1.7, 95%CI (-2.21:-0.42), P = 0.007), sham (-1.3±1.9, 95%CI (-2.28:-0.25), P = 0.018). Finally, heart rate was decreased after two weeks transcutaneous vagal nerve stimulation in comparison to sham treatment (-3.7 beats/min, 95%CI (-6.7:-0.6), P = 0.02).

## Conclusion

In this sham-controlled crossover study, we found no evidence that two weeks transcutaneous vagal nerve stimulation induces pain relief in patients with chronic pancreatitis.

## Trial registration number

The study is registered at NCT03357029; www.clinicaltrials.gov.

## Introduction

Chronic abdominal pain is the primary symptom of chronic pancreatitis (CP) and seen in the majority of patients during the course of disease [1]. Recent studies have provided evidence that CP patients have abnormal pain processing in the central nervous system [2]. Hence, sustained pancreatic nociceptive afferent inputs can over time result in sensitization of peripheral and central nociceptive pathways characterized by neuronal hyper-responsiveness. This can potentially lead to a chronic state of pain independent of peripheral nociceptive input, and patients typically respond inadequate to therapies directed against pathophysiological changes of the pancreas [1].

Vagal nerve stimulation (VNS) is a potential new non-pharmacological treatment targeting the autonomic and central nociceptive pathways. Previous studies have demonstrated that VNS provided analgesic effects in several painful conditions including fibromyalgia, pelvic pain and headache [3–5]. That has increased awareness of the potential of non-invasive vagal nerve stimulation (nVNS), and several medical devices are marketed. A study from our group, in healthy volunteers, showed that electrical and physiological modulation of parasympathetic tone resulted in pain attenuation [6], indicating that the parasympathetic nervous system is broadly anti-nociceptive [7]. nVNS has also shown its effect in clinical settings. Hence, two sham-controlled randomised controlled trials and six single-arm studies, demonstrated both rapid pain relief and a long term prophylactic effect of nVNS when used adjunctively with standard treatment in episodic and chronic migraine headaches [8]. The anatomical projections of the vagal nerve and its connections with many different brain areas involved in pain perception enables a potential analgesic effect [5]. The vagal nerve has widespread projections throughout the abdominal and pelvic viscera, thus a likely target of VNS could be abdominal pain in CP [9].

Based on the limited effectiveness of conventional treatments of CP pain, we hypothesized that two weeks nVNS could be effective as an adjuvant treatment to alleviate pain and improve quality of life in CP patients. The aims of this study were: 1) to primary evaluate the effects of two weeks nVNS on pain symptoms and quality of life by comparing nVNS treatment to sham treatment and 2) to investigate whether nVNS had an effect on modulating the autonomic nervous system. Finally, explorative analyses were provided comparing nVNS and sham treatment to their baselines.

## Materials and methods

### Study overview

This study was an investigator-initiated, randomised, double-blinded, sham-controlled, cross-over clinical trial assessing the effect of cervical nVNS for pain treatment in patients with CP (Fig 1). The study was conducted at Aalborg University Hospital, Denmark from January 2018 to April 2019. As active nVNS treatment, an FDA approved medical device (gammaCore-S, ElectroCore LLC, Basking Ridge, New Jersey, USA) was used, and the sham devices were identical in appearance (ElectroCore LLC, Basking Ridge, New Jersey, USA). The North Denmark Region Committee on Health Research Ethics (N-20170023) and the Danish Medical Agency (2017023686) approved the study. Furthermore, the study was registered at Clinicaltrials.gov (NCT03357029). This study was conducted in accordance with the Declaration of Helsinki, and consistent with Good Clinical Practice. For more information on the study design, please refer to the study protocol which has been published in BMJ Open [10] (S1 Appendix CON-SORT checklist and protocol).

### Inclusion of CP patients

Patients from the age of 18 years with a diagnosis of CP, based on the Mayo Clinic diagnostic criteria [11], were included regardless of the aetiology of CP. The included patients suffered from chronic abdominal pain more than 3 days per week lasting longer than 3 months, and they should consider their pain as insufficiently treated with usual analgesic treatments. All patients were required to read and understand Danish and to be able to comply with the scheduled visits and procedures. All participants signed and dated the informed consent and the Power of attorney document from the Danish Medical Agency [10]. Exclusion criteria were ongoing alcohol dependence, illegal drug dependencies, participating in other studies

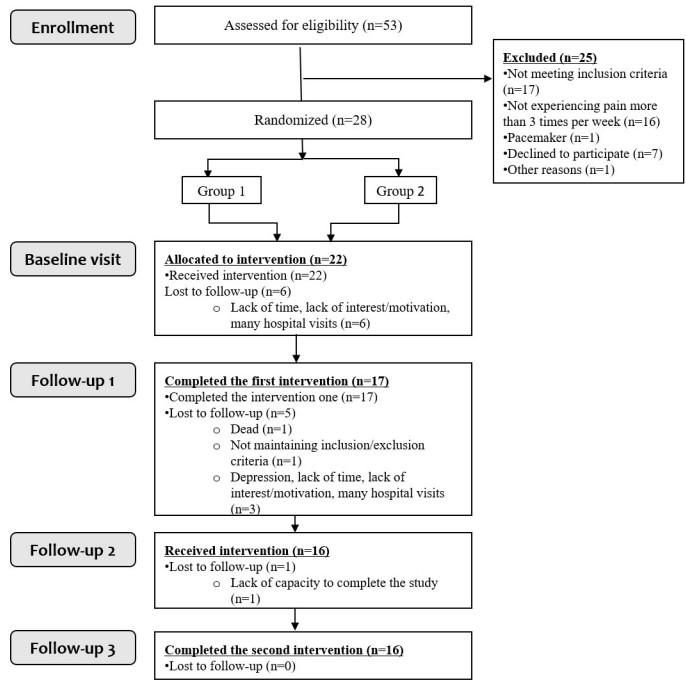

**Fig 1. CONSORT diagram.** Study enrolment and randomisation.

where an investigational drug was used. Patients with any clinically significant abnormalities that in the opinion of the investigator could increase the risk associated with trial participation were excluded. Moreover, patients with cardiovascular diseases, low blood pressure (<100/ 60), elevated intracranial pressure, female patients who were pregnant or lactating, contraindications for magnetic resonance imaging, previous surgery on the vagal nerve, and patients with known neuropathy were excluded. Patients were also excluded during the study, if they did not maintain the eligibility criteria during the study period [10]. All patients were asked to maintain their usual pain medication and dosage during the entire study, and they were only allowed to take extra pain medication in case of severe pain attacks. Any changes needed in pain medication during the study period was noted by the patient in the pain diary. Patients were allowed to continue the following analgesics during the study period: paracetamol, non-steroidal anti-inflammatory drug (NSAID), opioids, tricyclic antidepressant (TCA), serotonin-norepinephrine reuptake inhibitor (SNRI) and gabapentoids.

## Randomisation and blinding

Included patients were randomly assigned in two sequences (order of treatments: nVNS/sham or sham/nVNS) using block randomisation, allowing seven patients at the time to be randomised. The randomisation list was generated by an automatic web-based randomisation program. Patients and those administrating medical devices and assessing the outcomes were blinded to the group assignment.

## Interventions

The active treatment (nVNS) was performed using the medical device gammaCore-S, which is commercially available and FDA-approved for the treatment of primary headache [8]. The sham treatment was performed using a sham device. The two devices were identical in appearance, weight, visual and audible feedback, but the sham device did not deliver electrical stimuli (only a vibration). Patients were instructed to self-administer the treatment bilaterally on the neck three times per day for two weeks at home. Two weeks stimulation was chosen, since previous prospective, double-blind, sham-controlled, randomized studies with transcutaneous VNS have demonstrated that stimulations with gammaCore induced a pain reliving effect at 30, 60 and 120 minutes in migraine patients [12]. Therefore, we expected that two weeks would be sufficient to induce a long-term pain-relieving effect. The treatment was delivered via two stainless steel contact surfaces that were covered with conductive gel before each treatment (Fig 2). Next, the device was positioned in parallel with the carotid artery on the neck. One stimulation had a duration of two minutes [13]. On the first day before treatment, patients were trained on correct positioning and instructed to adjust the stimulation intensity using +/−buttons on the device to achieve a comfortable and sufficient sensation. The sufficient sensation was gained, when the patient step-by-step increased the intensity of the stimulation, which induced mild pain and unpleasantness. Then, the intensity was decreased one step immediately. The intensity of the stimulation varied from 1 to 40 units, and patients registered the intensity of the stimulation after each treatment in the pain diary. The treatment periods were separated by a two-week wash-out period. The patients continued their usual analgesic medication during the wash-out period. The two week wash-out period has been used in previous studies of neuromodulation and shown to be sufficient to reset the effects of neuromodulation [10].

Compliance was assessed by reading the remaining stimulations on the device display after each treatment period. Throughout both treatment periods, all patients were asked to register any adverse events using an adverse events questionnaire.

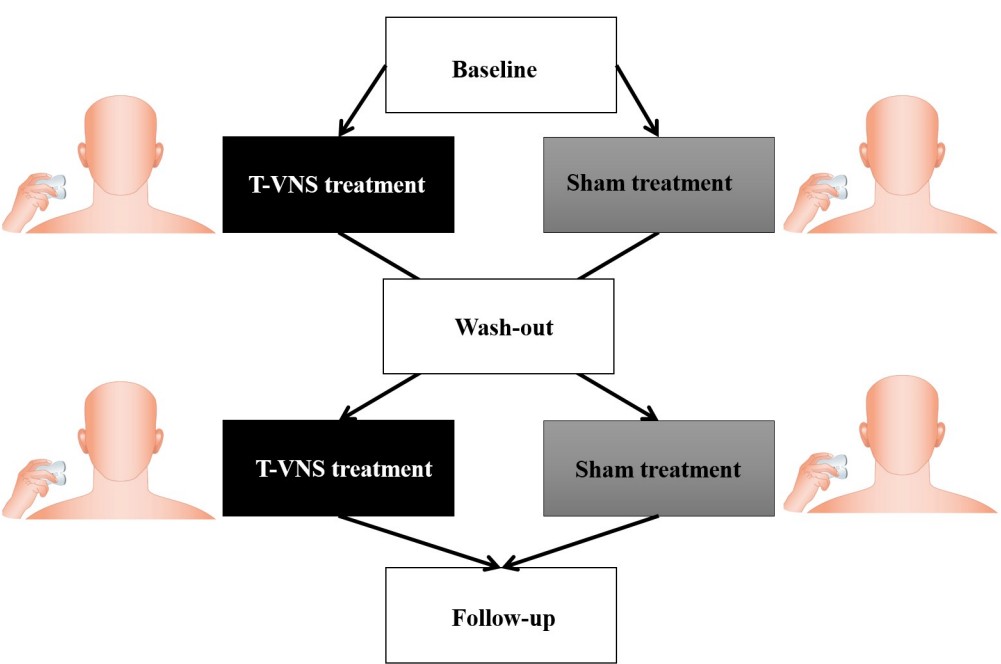

**Fig 2. Study design & treatment procedure: Patients were instructed to self-administer the treatment bilaterally on the neck three times per day for two weeks at home.** The treatment was delivered via two stainless steel contact surfaces that were covered with conductive gel before each treatment.

## Clinical assessment of pain and quality of life

The primary clinical endpoint was to evaluate the effects of two weeks nVNS on pain symptoms and quality of life compared to two weeks of sham treatment. The daily experience of pain (the maximum pain intensity and the average pain intensity) was measured using a pain diary based on a visual analogue scale (VAS), with 0: no pain, and 10: worst pain imaginable [14]. All patients were instructed to fulfil the pain diary every day during the entire 8-week study period, starting with baseline measurements for seven consecutive days before the first treatment. One baseline measurement and the effect of two weeks treatment on pain symptoms were measured as an average of the pain scores for the last four days of the baseline and treatment periods, respectively.

The secondary clinical endpoints included three questionnaires; 1) the European Organization for Research and Treatment of Cancer Quality of Life Questionnaire (EORTC QLQ-C30) [15], 2) Brief Pain Inventory (BPI) short form questionnaire [16], and 3) Patient's Global Impression of Change (PGIC) [17]. EORTC QLQ-C30 and BPI questionnaires were fulfilled before and after each treatment period, while the PGIC score was assessed at the end of each treatment period.

## Assessment of parasympathetic modulation effects

To investigate whether nVNS was able to modulate the cardiac vagal tone (CVT) in CP patients, the parasympathetic modulation was assessed using the electrocardiography (ECG) recorder device eMotion Faros 180 (Mega Electronics Ltd, Kuopio, Finland) [18]. First, the patients were instructed to rest for five minutes. Next, ECG electrodes (Ambu Blue Sensor P, Denmark) were placed in the left and right subclavicular areas and cardiac apex and then

attached to a non-invasive cardiac monitor. A biosignal acquisition system (ExtremeBio-metrics, London, UK) was used to assess cardiac derived parameters from five minutes record-ings. Recordings were conducted before and after two weeks of nVNS/sham treatment. Similarly, recordings were also conducted after two weeks of nVNS/sham treatment.

Based on the ECG recording, CVT was assessed as a measure of the influence of the para-sympathetic nervous system on the heart through the vagal nerve, which originates in the brain stem. CVT was calculated beat-to-beat based on detecting positive phase shifts in the R-R interval and used as a putative measurement of heart rate variability [18]. Thus, low CVT values reflect low heart rate variability, whereas high values reflect high heart rate variability. CVT was measured on a linear vagal scale (LVS) [18]. Before analysis, data were cleaned for noise, meaning that the first 10 samples of data were removed. Also, heart rate data points deviating from beat-to-beat with >15 beats pr. minute were removed, due to non-physiologi-cal causative factors. Since the consecutive R-R intervals affect the CVT measure, the previous and following 7 samples were also removed. Finally, heart rate was measured before and after both treatments.

## Statistical analyses

The study was powered to detect a minimal difference in average daily pain scores of 30% between nVNS and sham treatment. Based on a standard deviation (SD) of 40%, we deter-mined that a study with 16 patients in a crossover design was needed to provide a power of 80% with the use of a 2-sided significance level of 0.05 [10].

All data are presented as mean and SD unless otherwise stated. The continues variables were checked for normality using Shapiro-Wilk test. Before the primary analysis was per-formed, we defined the intensity of pain as an average of the scores for pain in the patient's diary for the last 4 days in each treatment period [19]. For analysis of the primary endpoint, the difference in pain intensities between the two treatment sequences (nVNS/sham and sham/nVNS) were calculated for each patient and compared using a t-test for independent samples. The assumption that the wash-out phase was long enough to rule out a carry-over effect was checked in a pretest by calculating the sum of pain intensity scores for each subject and comparing these across the two sequence groups by means of a t-test for independent samples [20]. Changes in secondary continues endpoints (BPI, EORTC QLQ-C30, CVT data) were analyzed using a similar approach. Data on compliance was transformed to a relative scale (%) and analysed using a paired sample t-test. Categorical data (PGIC and adverse events) were analyzed using a McNemar test. Finally, one-sample t-tests were used to further analyse the difference for 1) nVNS vs. baseline, and 2) sham vs. baseline (within group analy-sis). All the statistical analyses were performed using statistical software package STATA V.16.1 (StataCorp LP, College Station). P-values <0.05 were considered statistically significant.

## Results

### Patient enrolment and clinical characteristics

Fifty-three patients were screened and 28 underwent randomisation (Fig 1). Twenty-two patients completed the baseline assessment. 17 patients completed the first treatment period and 16 patients completed both treatment periods. Demographic and clinical characteristics of the enrolled patients (n = 22) and for patients completing the study (n = 16) are summarized in Table 1. Characteristics for patients in each sequence (nVNS/sham, n = 4 and sham/nVNS, n = 12) are provided in Supporting information (S1 Table: Demographics by sequences). Five out of six dropouts were assigned to the nVNS/sham sequence. No patients dropped out due

Table 1. Demographic and clinical characteristics of chronic pancreatitis patients.

| Variables | Chronic pancreatitis patients | |
|---|---|---|
| | Included in study (n = 22) | Completed study (n = 16) |
| Age (years) | 55.2±8.9 | 56.6±9.4 |
| Male, n (%) | 18 (81.8) | 14 (88) |
| Body mass index (kg/m$^2$) | 22.5±4.6 | 21.9±3.6 |
| **Aetiology of chronic pancreatitis, n (%)** | | |
| Alcohol | 13 (35) | 11 (42) |
| Nicotine | 15 (41) | 10 (38) |
| Nutritional | 0 (0) | 0 (0) |
| Hereditary factors | 3 (8) | 2 (8) |
| Efferent duct factors | 5 (13) | 3 (12) |
| Immunological factors | 0 (0) | 0 (0) |
| Miscellaneous and rare metabolic factors | 1 (3) | 0 (0) |
| Duration of chronic pancreatitis (years) | 8.1±7.6 | 9.7±8.4 |
| Diabetes, n (%) | 11 (50) | 9 (56) |
| **Analgesics, n (%)** | | |
| Paracetamol | 14 (64) | 11 (36) |
| NSAID | 2 (9) | 1 (3) |
| Opioids | 19 (86) | 13 (42) |
| **Adjuvant analgesics, n (%)** | 1 (5) | 1 (3) |
| TCA | 0 (0) | 0 (0) |
| SNRI Gabapentoids | 7 (32) | 5 (16) |
| **Current alcohol use, n (%)** | | |
| 0 units | 12 (55) | 8 (50) |
| 1–5 units | 5 (23) | 3 (19) |
| 5–10 units | 5 (23) | 5 (31) |
| 10–15 units | 0 (0) | 0 (0) |
| >20 units | 0 (0) | 0 (0) |
| **Current smoking pattern, n (%)** | | |
| non-smoker | 5 (23) | 4 (25) |
| 1–5 cigarettes per day | 3 (14) | 2 (13) |
| 5–10 cigarettes per day | 2 (9) | 2 (13) |
| 10–15 cigarettes per day | 3 (14) | 2 (13) |
| 15–20 cigarettes per day | 6 (27) | 3 (19) |
| >20 cigarettes per day | 3 (14) | 3 (19) |

Note: Values are means ± SD. Percentages may not total 100 due to rounding.

Abbreviations: n = number of patients. NSAID = Nonsteroidal anti-inflammatory drug. TCA = Tricyclic antidepressant. SNRI = Serotonin-norepinephrine reuptake inhibitor.

to adverse effects or non-tolerance of the interventions. In the nVNS treatment, 98%±3% of all stimulations were applied compared with 94%±11% in the sham treatment (P = 0.24). Furthermore, the average intensity of the stimulations was 31.8±6.9 units during nVNS treatment and 36.0±7.9 units for sham treatment (P = 0.02).

## Primary and secondary clinical outcomes

**nVNS vs. sham.** Primary and secondary clinical endpoints (after two weeks intervention) are summarized in Table 2. No significant differences were observed for the maximal or

**Table 2. Primary and secondary clinical endpoints after two weeks of nVNS and sham treatment.**

| Primary clinical endpoints | Pain diary | | | | | | | |
|---|---|---|---|---|---|---|---|---|
| | Variables | Sequence | Treatment period Mean ±SD | | Within-individual differences for nVNS vs. sham | Treatment effect for nVNS vs. sham (95% CI) | P (effect) | P (carry-over) |
| | | | 1 | 2 | | | | |
| | Average pain score (VAS) | nVNS/sham | 2.9±1.8 | 2.2±2.1 | 0.34±0.61 | 0.17 (-0.86:1.20) | 0.72 | 0.49 |
| | | sham/nVNS | 3.6±2.5 | 3.3±2.4 | -0.17±0.88 | | | |
| | Maximal pain score (VAS) | nVNS/sham | 4.1±2.3 | 3.4±2.7 | 0.34±0.77 | 0.20 (-1.14:1.54) | 0.76 | 0.56 |
| | | sham/nVNS | 4.6±2.6 | 4.4±2.5 | -0.15±1.15 | | | |
| Secondary clinical endpoints | Brief Pain Inventory questionnaire | | | | | | | |
| | BPI Pain | nVNS/sham | 3.1±2.1 | 2.8±2.5 | 0.1±0.4 | 0.19 (-1.11:1.48) | 0.76 | 0.40 |
| | | sham/nVNS | 3.9±2.2 | 4.0±2.4 | 0.1±1.2 | | | |
| | BPI Interference | nVNS/sham | 2.0±1.2 | 2.1±1.1 | -0.1±0.2 | -0.006 (-1.17:1.15) | 0.99 | 0.13 |
| | | sham/nVNS | 3.8±2.1 | 3.9±2.6 | 0.0±1.0 | | | |
| | Quality of life | | | | | | | |
| | Global health status | nVNS/sham | 64.6±19.7 | 66.7±22.6 | -1.0±4.0 | -1.04 (-13.1:11.0) | 0.86 | 0.13 |
| | | sham/nVNS | 50.7±15.7 | 50.7±20.2 | 0.0±10.8 | | | |
| | Physical functioning | nVNS/sham | 86.7±34.9 | 78.3±14.8 | 4.2±17.3 | 4.44 (-7.3:16.2) | 0.43 | **0.04** |
| | | sham/nVNS | 59.4±18.1 | 60.0±17.3 | 0.3±5.8 | | | |
| | Role functioning | nVNS/sham | 83.3±16.7 | 79.2±16.0 | 0.0±8.3 | 3.5 (-7.2:14.1) | 0.49 | 0.08 |
| | | sham/nVNS | 50.0±24.6 | 56.9±28.8 | 3.5±7.5 | | | |
| | Emotional functioning | nVNS/sham | 81.3±14.2 | 81.3±14.2 | 0.0±3.4 | -1.7 (-13.5:10.0) | 0.76 | 0.09 |
| | | sham/nVNS | 65.3±20.4 | 61.8±21.2 | -1.7±10.6 | | | |
| | Cognitive functioning | nVNS/sham | 83.3±19.2 | 87.5±16.0 | -2.1±4.2 | -2.1 (-12.7:8.5) | 0.68 | **0.02** |
| | | sham/nVNS | 55.6±25.9 | 55.6±17.9 | 0.0±9.4 | | | |
| | Social functioning | nVNS/sham | 83.3±19.2 | 87.5±16.0 | -2.1±4.2 | -4.2 (-16.2:7.9) | 0.47 | 0.15 |
| | | sham/nVNS | 66.7±23.6 | 62.5±30.3 | -2.1±10.7 | | | |
| | Fatigue | nVNS/sham | 29.6±17.0 | 22.2±18.1 | 7.4±3.2 | 6.5 (-7.5:20.5) | 0.34 | 0.16 |
| | | sham/nVNS | 45.4±20.9 | 43.5±30.9 | -0.9±10.8 | | | |
| | Nausea and vomiting | nVNS/sham | 0.0±0.0 | 0.0±0.0 | 0.0±0.0 | 1.4 (-8.0:10.8) | 0.76 | 0.19 |
| | | sham/nVNS | 5.6±8.2 | 8.3±16.7 | 1.4±8.6 | | | |
| | Pain | nVNS/sham | 45.8±25.0 | 33.3±13.6 | 6.2±8.0 | 4.2 (-13.1:21.4) | 0.61 | 0.12 |
| | | sham/nVNS | 59.7±21.9 | 55.6±25.9 | -2.1±15.1 | | | |
| | Dyspnoea | nVNS/sham | 0.0±0.0 | 16.7±19.2 | -5.6±9.6 | -6.9 (-24.7:10.8) | 0.41 | 0.28 |
| | | sham/nVNS | 22.2±25.9 | 19.4±26.4 | -1.4±13.2 | | | |
| | Insomnia | nVNS/sham | 11.1±19.2 | 16.7±33.3 | 5.6±9.6 | 6.9 (-25.7:39.5) | 0.65 | 0.06 |
| | | sham/nVNS | 38.9±37.2 | 41.7±37.9 | 1.4±25.1 | | | |
| | Appetite loss | nVNS/sham | 22.2±19.2 | 8.3±16.7 | 11.1±9.6 | 13.9 (0.5:27.3) | **0.04** | 0.37 |
| | | sham/nVNS | 27.8±34.3 | 33.3±37.6 | 2.8±9.6 | | | |
| | Constipation | nVNS/sham | 0.0±0.0 | 0.0±0.0 | 0.0±0.0 | 0.0 (-15.6:15.6) | 1.00 | 0.29 |
| | | sham/nVNS | 13.9±30.0 | 13.9±26.4 | 0.0±14.2 | | | |
| | Diarrhoea | nVNS/sham | 16.7±19.2 | 8.3±16.7 | 4.2±8.3 | -2.8 (-15.9:10.4) | 0.66 | 0.22 |
| | | sham/nVNS | 41.7±32.2 | 27.8±37.2 | -6.9±11.1 | | | |
| | Financial difficulties | nVNS/sham | 16.7±19.2 | 25.0±16.7 | -4.2±8.3 | 0.0 (-12.3:12.3) | 1.00 | 0.29 |
| | | sham/nVNS | 30.6±22.3 | 38.9±27.8 | 4.2±10.4 | | | |
| Secondary experimental endpoints | Cardiac-derived parameters | | | | | | | |
| | Cardiac vagal tone (LVS) | nVNS/sham | 6.1±3.8 | 5.3±4.5 | 0.4±0.7 | 0.5 (-0.3:1.2) | 0.18 | 0.48 |
| | | sham/nVNS | 4.6±1.9 | 4.7±1.9 | 0.1±0.5 | | | |
| | Heart rate (beats/min) | nVNS/sham | 65.5±20.1 | 72.3±20.8 | -3.4±2.8 | -3.66 (-6.7:-0.6) | **0.02** | 0.39 |
| | | sham/nVNS | 63.5±8.5 | 62.9±5.3 | -0.3±2.4 | | | |

Note: Comparisons between nVNS and sham treatments for the two treatment periods shown by treatment sequence (nVNS/sham, n = 4 and sham/nVNS, n = 12). The difference in pain intensities between the two treatment sequences were calculated for each patient and compared using a t-test for independent samples. One patient had missing data for role functioning, fatigue, dyspnoea, insomnia and appetite loss and one patient had missing data for cardiac vagal tone.

Abbreviations: VAS = visual analogue scale. SD = standard deviation. CI = confidence interval. nVNS = non-invasive vagal nerve stimulation. LVS = linear vagal scale. N = number of patients.

average pain score between two weeks nVNS treatment vs. sham treatment (difference in average pain score (visual analogue scale): 0.17, 95%CI (-0.86;1.20), P = 0.7) (Fig 3 and Supporting information (S1 Fig: Individual plots of the maximal and average pain scores)). Likewise, no differences were seen for secondary clinical endpoints, except from a significant increase in

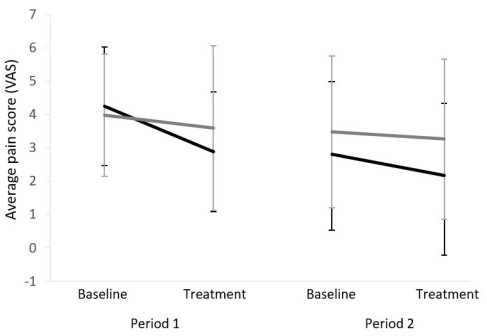

**Fig 3. Average pain score during the two study periods for the two sequences (nVNS/sham in black and sham/nVNS in grey).** VAS = visual analogue scale.

the appetite loss score with higher scores observed after nVNS (13.9, 95%CI (0.5:27.3), P = 0.04). PGIC showed no significant changes for nVNS treatment compared with sham treatment (Table 3, P = 0.63). The primary and secondary clinical endpoints with correction for the baseline assessments are given in Supporting information (S2 Table: The primary and secondary clinical endpoints corrected for baseline assessments), where no changes in pain scores were observed; however, a treatment effect of nVNS was shown with improvements in physical functioning (14.6, 95%CI (1.3:28.0), P = 0.03) and dyspnoea (-19.4, 95%CI (-37.3:1.6), P = 0.04).

**nVNS and sham vs. baseline (within group analysis).** A significant reduction was found in the average pain scores (-0.7±1.2, 95%CI (-1.31:-0.003), P = 0.049) and the maximal pain score (-1.3±1.7, 95%CI (-2.21:-0.42), P = 0.007) comparing nVNS treatment with baseline. Also, significant reductions were detected in the maximal pain scores (-1.3±1.9, 95%CI (-2.28:-0.25), P = 0.018) and trend towards a reduction in the average pain scores (-0.6±1.2, 95%CI (-1.19:0.03), P = 0.06) for sham treatment as compared with baseline.

## Secondary experimental outcomes

Cardiac-derived parameters are reported in Table 2. Compared to sham, nVNS treatment induced a decrease in heart rate (-3.7 BPM, 95% CI (-6.7:-0.6), P = 0.02), while no change in the CVT was observed (P = 0.18). Cardiac-derived parameters with correction for the baseline assessments showed no significant differences between the nVNS and sham treatments (Supporting information S2 Table: The primary and secondary clinical endpoints corrected for baseline assessments).

## Adverse events

Five serious adverse events (SAEs) occurred during the study. One patient had a SAE with hospitalization due to acute pancreatitis in the nVNS period. Another patient had a SAE with death during the sham treatment, but this was not related to the intervention. Two patients had a SAE with hospitalization during the wash-out period (due to worsening of pain). During the follow-up period, a patient was admitted due to a skin rash. All SAEs were considered to be unrelated to the nVNS or sham treatment. There was no significant difference between the nVNS and sham treatment on any of the adverse events. Detailed information on adverse events are provided in Table 4.

**Table 3. Patient's Global Impression of Change (PGIC).**

| PGIC, n | | nVNS | | | P(exact) |
|---|---|---|---|---|---|
| | | None | Yes | | 0.63 |
| **Sham** | None | 8 | 1 | 9 | |
| | Yes | 3 | 4 | 7 | |
| | | 11 | 5 | 16 | |

Note: Distribution of improvement in PGIC induced by the nVNS and sham treatments (None: No change or worse; Yes: Improved). Abbreviations: n = number of patients.

**Table 4. Adverse events.**

| **Nausea, n** | | **nVNS** | | | **P(exact)** |
|---|---|---|---|---|---|
| | | None | Yes | | 0.50 |
| **Sham** | None | 1 | 0 | 1 | |
| | Yes | 2 | 13 | 15 | |
| | | 3 | 13 | 16 | |
| **Constipation, n** | | **nVNS** | | | **P(exact)** |
| | | None | Yes | | 1.00 |
| **Sham** | None | 1 | 0 | 1 | |
| | Yes | 0 | 15 | 15 | |
| | | 1 | 15 | 16 | |
| **Vomit, n** | | **nVNS** | | | **P(exact)** |
| | | None | Yes | | 0.50 |
| **Sham** | None | 0 | 0 | 0 | |
| | Yes | 2 | 14 | 16 | |
| | | 2 | 14 | 16 | |
| **Dizziness, n** | | **nVNS** | | | **P(exact)** |
| | | None | Yes | | 0.63 |
| **Sham** | None | 1 | 1 | 2 | |
| | Yes | 3 | 11 | 14 | |
| | | 4 | 12 | 16 | |
| **Headache, n** | | **nVNS** | | | **P(exact)** |
| | | None | Yes | | 1.00 |
| **Sham** | None | 0 | 2 | 2 | |
| | Yes | 2 | 12 | 14 | |
| | | 2 | 14 | 16 | |
| **Fatigue, n** | | **nVNS** | | | **P(exact)** |
| | | None | Yes | | 1.00 |
| **Sham** | None | 3 | 1 | 4 | |
| | Yes | 2 | 10 | 12 | |
| | | 5 | 11 | 16 | |
| **Abdominal pain, n** | | **nVNS** | | | **P(exact)** |
| | | None | Yes | | 0.38 |
| **Sham** | None | 1 | 1 | 2 | |
| | Yes | 4 | 10 | 14 | |
| | | 5 | 11 | 16 | |

(*Continued*)

**Table 4.** (Continued)

| Nausea, n | | nVNS | | | P(exact) |
|---|---|---|---|---|---|
| **Diarrhoea, n** | | **nVNS** | | | **P(exact)** |
| | | None | Yes | | 1.00 |
| **Sham** | None | 1 | 1 | 2 | |
| | Yes | 2 | 12 | 14 | |
| | | 3 | 13 | 16 | |
| **Dry mouth, n** | | **nVNS** | | | **P(exact)** |
| | | None | Yes | | 1.00 |
| **Sham** | None | 2 | 0 | 2 | |
| | Yes | 1 | 13 | 14 | |
| | | 3 | 13 | 16 | |
| **Itching, n** | | **nVNS** | | | **P(exact)** |
| | | None | Yes | | 1.00 |
| **Sham** | None | 0 | 1 | 1 | |
| | Yes | 0 | 15 | 15 | |
| | | 0 | 16 | 16 | |
| **Other, n** | | **nVNS** | | | **P(exact)** |
| | | None | Yes | | 0.25 |
| **Sham** | None | 0 | 0 | 0 | |
| | Yes | 3 | 13 | 16 | |
| | | 3 | 13 | 16 | |

Note: Distribution of specific adverse events after nVNS and sham treatment (None: No adverse events; Yes: Adverse events noted).

Abbreviations: n = number of patients.

## Discussion

To the best of our knowledge, this is the first clinical trial in CP patients evaluating the effects of nVNS on chronic abdominal pain. Overall, we found no evidence of significant difference in pain relief between the two treatments. A significant increase was demonstrated for appetite loss. However, both the nVNS and sham treatments induced significant improvements in pain scores when compared to their baselines. An enhanced parasympathetic tone was likely induced after two weeks nVNS, as reflected as a decrease in heart rate.

### Effect of nVNS in patients with chronic pain

Previous clinical and pre-clinical studies of VNS provided strong evidence of analgesic effects in various clinical conditions including patients with fibromyalgia, pelvic pain and headaches [3]. In contrast to our hypothesis, we found no evidence that CP patients had a significant pain reduction comparing the nVNS and sham treatments. Interestingly, we found a reduction in pain when comparing nVNS treatment with baseline, and the same was found for the sham treatment. Previous clinical studies of pain treatment in CP patients have also shown an extensive placebo response [21] not only in pharmacological studies, but also in studies with surgical interventions, showing an improvement in subjective outcomes, including pain, disability and quality of life [22,23]. Similar pattern was observed in the current study, showing a potential sham effect in CP patients. A large pain reduction in the group receiving sham may mask the

genuine efficacy of nVNS, and this may explain the non-significant results when comparing nVNS and sham treatment.

The lack of an anti-nociceptive effect following nVNS treatment, compared with sham treatment, could be explained by only partial enhancement of the parasympathetic nervous system to evoke anti-nociceptive effects as we only found an effect on the heart rate but not in CVT. The lack of pain improvement may also reflect that a longer-lasting treatment is needed, but the study could also be under-powered to detect a potential effect. Although, our findings are inconsistent with other VNS studies where sham control was not performed, it is important to emphasize that the influences of confounding covariates were reduced in our study as the patients serve as their own controls. As far as we are aware, the majority of the VNS studies were open labelled and non-crossover studies [24–27]. A study by Lange et al., provided evidence that VNS may be a useful adjunct treatment for patients with fibromyalgia [24], and Oshinsky et al. demonstrated that pain reduction was correlated with a reduction in the extracellular glutamate concentration in the brain [28]. Compared to these non-crossover studies, our current study found similar results, i.e. pain reduction comparing nVNS treatment with baseline. Our negative findings could therefore likely be explained by the strong study-design (double-blinded sham-controlled crossover study).

Recently, a functional MRI study showed that cervical vagal afferents can be accessed non-invasively via transcutaneous electrical stimulation of the neck, which overlies the course of the vagus nerve [29]. Also, the brain activation elicited by nVNS was analogous to that elicited by invasive VNS, providing validation of cervical nVNS effects [27].

## Modulation of the parasympathetic tone with nVNS in CP patients

A previous study by Brock et al., demonstrated that nVNS induced an increase in CVT and reduced serum TNF-α in healthy volunteers, suggesting that nVNS exerts an autonomic and anti-inflammatory effect [30]. Thus, we expected that CVT would significantly increase in CP patients after two-week nVNS treatment as compared with sham treatment. However, we only found a decrease in heart rate after two weeks nVNS treatment as compared with sham, but no change in CVT. This indicates that nVNS might only be partially modulating the parasympathetic tone in CP patients. This could be explained by low sample size, as the study was powered for the primary endpoint (pain scores) and not for CVT. A study by Juel et al. in CP patients, showed that nVNS and deep slow breathing increased CVT compared to sham stimulation [6]. In this study, also no effects on pain were found. However, caution must be applied when comparing studies, as Juel et al. measured the acute effect of auricular VNS in combination with deep slow breathing implying that nVNS alone may not be adequate to modulate the vagal tone in CP patients.

Despite unsuccessful enhancement of mean CVT in our study, nVNS could potentially be able to module the parasympathetic tone in a sub-group of patients. A study conducted by Farmer et al., demonstrated that the 95% confidence interval of CVT was 1.9–17.8 LVS for healthy persons [18]. The average CVT in our study was in the lower part of the normal range, indicating that it is likely not possible to enhance the CVT in all patients. Despite one of the exclusion criteria was known peripheral neuropathy, some patients could have autonomic neuropathy. Hence, some patients could have vagal neuropathy, which can make it more difficult to modulate the parasympathetic tone and potentially mediating an anti-hyperalgesic effect [7]. A study by Liu et al. found that epilepsy patients having reduced heart rate viability compared to healthy controls, had significantly lower effect of VNS treatment (non-responders) [31]. Thus, the effect of nVNS could be related to the degree of autonomic dysfunction, and CVT may be useful as a screening tool in future studies. Accumulating evidence suggests

that the autonomic nervous system plays an important role in the modulation of pain [7]. Considering that the parasympathetic nervous system acts broadly anti-nociceptive, while the sympathetic nervous system acts mainly pro-nociceptive, it is implicated that a balance between them is required for normal pain perception [32]. Thus, lack of an increase in the parasympathetic tone may lead to lack of prevention of visceral pain hypersensitivity in CP patients [7]. Additionally, an imbalance between the parasympathetic and sympathetic nervous systems may represent an important pathophysiological factor in CP patients with visceral pain. However, the latter should be investigated in detail in future studies.

## Limitations

A number of study limitations need to be considered when evaluating our results. First, even though a sample size calculation was performed prior to this study, such calculations in clinical studies with experimental interventions are often based on some uncertain assumptions. Hence, we could have a too small sample with a risk of false negative results regarding our primary outcomes. Second, the paradigm and parameters of nVNS may influence the effect of the treatment including treatment duration, number of daily stimulations, stimulation intensity, unilateral vs. bilateral stimulation, etc. In this present study, the duration of the treatment was two weeks. There is no consensus regarding the duration of the treatment, and previous studies have used an intervention duration ranging from two weeks to six months in order to induce pain relieving effects [3,26–28,33]. The duration of treatment seems to differ depending on the type of disease, type of intervention (invasive VNS, auricular or cervical nVNS), and time frame (short-term vs. long-term effect) [3]. For example, in chronic pain patients, an auricular nVNS study applied the treatment for six weeks [34], while a cervical nVNS study applied the treatment for two weeks [12]. Given that CP is characterized by long-lasting ailment, future studies should possibly include a longer treatment duration to recover the functional plasticity in neural circuits of pain processing. In addition, the duration of the wash-out period of two weeks was chosen since it has been shown to be sufficient to reset the effects of neuromodulation [10], but long-term effects have also been demonstrated in patients with epilepsy [35]. However, if we extended the study duration (treatment and wash-out periods), a high risk of dropout would potentially occur, and study completion would become more difficult. Third, even though the sham device did not deliver electrical stimulation, it may not be physiologically inert as it provides some vibration and tactile sensation that potentially can stimulate the vagal nerve and induce an effect similar to the active treatment. This could also explain why there was no differences between the active and sham treatment, but a positive effect when comparing both the active and sham treatments to their respective baselines. Additionally, the order of treatment sequence was, despite block randomisation, unbalanced with more dropouts in the nVNS/sham sequence. It is unclear how this could have affected our findings, even though the statistical analysis approach takes the treatment sequence into account. Forth, in the current study, we did not analyse blood samples measuring inflammation. Although, no pain reductions were observed with two weeks nVNS, it would be interesting to assess whether the pancreatic inflammation was decreased. Fifth, even though the study was based on a cross-over design, where patients served as their own control, bias could be present due to the inclusion/exclusion criteria. For instance, ongoing alcohol dependence was an exclusion criteria whereas ongoing smoking was not, but nicotine could potentially impact the inflammatory-system [36]. Finally, various confounding factors are present in this patient cohort. For instance, the patients used several types of analgesics which may influence the effect of nVNS and as already discussed it may be difficult to modulate the parasympathetic tone for some patients. Furthermore, the impact of concomitant medication, the duration of

CP or other disease characteristics (including stress, anxiety, and depression) could also be relevant to investigate. In future studies, a larger sample size is also required in order to investigate sub-groups of patients, who possible could have an effect of nVNS.

## Conclusion

Our sham-controlled crossover study provided no evidence that adjuvant treatment with two weeks nVNS induces pain relief as compared to sham treatment in patients with painful chronic pancreatitis. Similarly, pain interference and quality of life scores were not improved in response to nVNS as compared with sham treatment. However, attention should be directed towards a potential pronounced sham effect of neuromodulation, since improvements were seen for both active and sham treatments as compared to baseline.

## Supporting information

**S1 Fig. Individual plots of the maximal and average pain scores.**
(DOCX)

**S1 Table. Demographics by sequences.**
(DOCX)

**S2 Table. The primary and secondary clinical endpoints corrected for baseline assessments.**
(DOCX)

**S1 Appendix. CONSORT checklist and protocol.**
(PDF)

## Acknowledgments

Electrocore LLC provided the gammaCore devices but did not influence the design or reporting of the study. Statistician Niels Henrik Bruun, MSc, at Unit of Clinical Biostatistics, Aalborg University Hospital, Aalborg, Denmark is acknowledged for support with the statistical analyses of the primary and secondary endpoints. We thank radiographer Kenneth Krogh Jensen at Department of Radiology, Aalborg University Hospital, Aalborg Denmark for his assistance in data collection.

## Author Contributions

**Conceptualization:** Janusiya Anajan Muthulingam, Søren Schou Olesen, Tine Maria Hansen, Christina Brock.

**Data curation:** Janusiya Anajan Muthulingam, Jens Brøndum Frøkjær.

**Formal analysis:** Janusiya Anajan Muthulingam, Søren Schou Olesen, Tine Maria Hansen, Jens Brøndum Frøkjær.

**Funding acquisition:** Jens Brøndum Frøkjær.

**Investigation:** Søren Schou Olesen, Asbjørn Mohr Drewes, Jens Brøndum Frøkjær.

**Methodology:** Søren Schou Olesen, Tine Maria Hansen, Christina Brock, Asbjørn Mohr Drewes, Jens Brøndum Frøkjær.

**Project administration:** Janusiya Anajan Muthulingam, Asbjørn Mohr Drewes, Jens Brøndum Frøkjær.

**Supervision:** Asbjørn Mohr Drewes, Jens Brøndum Frøkjær.

**Writing – original draft:** Janusiya Anajan Muthulingam.

**Writing – review & editing:** Søren Schou Olesen, Tine Maria Hansen, Christina Brock, Asbjørn Mohr Drewes, Jens Brøndum Frøkjær.

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
