## [Decision Letter · Decision Letter 0]

21 Jul 2020

PONE-D-20-13163

The effects of two weeks transcutaneous vagal neuromodulation in chronic pancreatitis patients: a randomised sham controlled clinical trial

PLOS ONE

Dear Dr. Frøkjær,

Thank you for submitting your manuscript to PLOS ONE. After careful consideration, we feel that it has merit but does not fully meet PLOS ONE’s publication criteria as it currently stands. Therefore, we invite you to submit a revised version of the manuscript that addresses the points raised during the review process.

Specifically, the reviewers had multiple comments and concerns about methods reporting, methods justification, and would like several points addressed in the discussion.

We look forward to receiving your revised manuscript.

Kind regards,

Susan Hepp

Academic Editor

PLOS ONE

Journal Requirements:

Additional Editor Comments (if provided):

Reviewers' comments:

Reviewer's Responses to Questions

**Comments to the Author**

1. Is the manuscript technically sound, and do the data support the conclusions?

Reviewer #1: Yes

Reviewer #2: Yes

Reviewer #3: Partly

2. Has the statistical analysis been performed appropriately and rigorously? 

Reviewer #1: Yes

Reviewer #2: Yes

Reviewer #3: No

3. Have the authors made all data underlying the findings in their manuscript fully available?

Reviewer #1: Yes

Reviewer #2: Yes

Reviewer #3: No

4. Is the manuscript presented in an intelligible fashion and written in standard English?

Reviewer #1: Yes

Reviewer #2: Yes

Reviewer #3: Yes

5. Review Comments to the Author

Reviewer #1: The study describes effects of the cervical transcutaneous vagus nerve stimulation in chronic pancreatitis patients. Irrespective of negative findings, this study design and results are of high value for the interested reader.

Ad Title

„Transcutaneous vagal neuromodulation“ is too general, leading potentially to confusion with the auricular transcutaneous vagal neuromodulation. I would recommend to integrate either “cervical” or “on the neck”

Ad Abstract

Please add to the methodology section “cervical” or “on the neck”. Currently the reader of the abstract has no clue where the stimulation was performed.

Add Abbreviation

The abbreviation t-VNS is a bit misleading since tVNS is usually used for the transcutaneous auricular stimulation, just in contrast to pVNS as used for the percutaneous auricular stimulation. The stimulation on the neck is usually abbreviated as nVNS. It is a suggestion to avoid confusion of the readership.

Ad Methods

I think that 2 weeks stimulation is a relatively short period, especially in view of a long-lasting chronic ailment with the associate changes in plasticity. For instance, the auricular VNS is usually applied for 6 weeks in chronic ailments such as the chronic back pain (https://doi.org/10.1213/01.ane.0000107941.16173.f7). Even though it is shortly discussed in the limitations section, the reader would welcome a more detailed discussion on the effective treatment durations of VNS and cervical transcutaneous VNS (optionally, including also auricular VNS) in chronic ailments. Please discuss and justify this issue in detail.

Please amend whether state-of-the-art individual treatment of patients was optimized before the treatment, interrupted or continued during the tVNS study. Address concomitant medication/treatment of patients which is very important since improvements were registered with respect to the baseline.

Quantitative description of CVT is missing, even though a reference is given. Please describe it in a few sentences so that the reader has a clear idea about it.

More information is needed on the wash-out period, how long, 1 week? With uninterrupted concomitant treatment?

Ad Discussion

p.15 Discussed comparison/justification of cervical tVNS results with respect to invasive VNS results is limited due to the basically different technologies applied. Therefore, please address potential validation of cervical tVNS effects with respect to invasive VNS effects to substantiate your conclusions here.

Reviewer #2: In the present study, the authors used the antinociceptive effect of the vagus nerve (VN) to perform a two weeks non-invasive transcutaneous cervical VN stimulation (VNS) in patient with chronic pancreatitis (CP), in a sham-controlled crossover study. Unfortunately, VNS did not induce pain relief in patients with CP.

This is an interesting and original study as stated by the authors, since there are no clinical trials published on VNS in CP. I already had the opportunity to review the study protocol published in BMJ Open in 2019 by the same authors (ref. 10).

There are some points which need to be addressed:

- Abstract: specify that it was a cervical transcutaneous VNS. Specify the device (GammaCore).

- The number of patients is rather small for me, as specified by the authors.

- VNS stimulation is generally performed on the left VN while in the present study the authors stimulated both VN. Is-there any advantage to perform such a bilateral stimulation?

- The duration of the study (2 weeks of stimulation) is most likely too short since VNS is a slow acting therapy as demonstrated for epilepsy. Indeed, VNS induces modifications of neuronal plasticity, which are progressive. CP is not comparable to “headache” which is FDA approved with the device GammaCore. As stated by the authors CP induces central modifications which can induce a central sensitivity syndrome. In addition, most of these patients use opioid compounds which are also able to induce central perturbation of pain. So, it is simplistic to think that a two-week VNS will be able to improve such patients. In addition, if the effect of t-VNS is delayed over time, when switching to sham stimulation, it could induce a bias of the results of sham stimulation.

- In addition, the sham stimulation is rather questionable since it induces a vibration which, as noted by the authors, is able to have an effect on the auricular branch of the VN.

- The authors did not discuss the parameters of VNS.

- The duration of CP is 8.1 ± 7.6 years, so very variable. Thus, the central integration of pain and central sensitization is supposed to be patient-dependent and different between patients.

- The analgesic treatments may interfere with VNS (see above)

- Pain is multidimensional and rather subjective, so the use of a VAS is questionable, although I agree that VAS is often used in such conditions. It seems to me that there are no questionnaires evaluating stress (perceived), depression, anxiety, coping strategies in this trial.

- Did the patients used or have used complementary medicines which are able to impact the pain factor?

- A limit for me of non-invasive VNS by comparison to invasive one is the compliance. How the authors controlled that patients had regularly performed VNS (and bilaterally)?

- Regarding vagal tone, did the authors include patients with only low vagal tone which should be more appropriate and is supposed to better beneficiate of VNS. It is not clear for me in the manuscript.

- The authors did not explore the anti-inflammatory effect of VNS which is well known in different chronic inflammatory conditions such as rheumatoid arthritis and Crohn’s disease. That would have been interesting and there is a relation between inflammation and pain. This point is only stated in the “Limitations” of the study.

- Ongoing alcohol dependence was an exclusion criterion but not smoking? Nicotine is able to act on alpha7 nicotinic acetylcholine receptor and to have an anti-inflammatory effect.

Reviewer #3: *** General remarks: ***

The study follows a two period two treatment crossover design.

However, the analysis which was performed is too simplistic and fails to account adequately for the actual trial design.

The text itself is marred somewhat by a variety of minor syntactical, grammatical, and English usage errors. Since the authors did not provide page and line numbers as requested by PLOS ONE submission guidelines, no specific corrections will be offered.

*** Specific remarks: ***

The usual modern analysis for a crossover design involves using a linear mixed effects model with fixed effects for period and treatment, and random effects for sequence and subject nested in sequence. The analysis of variance results should be presented, as well as effect estimates. Consult Senn (2000) for the details.

For endpoints that the authors wish to model in finer detail, such as those involving measurements taken over time during each period, the authors may consider either summarizing the endpoints or elaborating the linear mixed effects model with within-period measurements. Summarizing is a less satisfactory method, because inference is then only to the summary measures rather than to the actual assessments of interest. The dependence due to repeated measurements over time should be accounted using an appropriate R-side correlation structure with either an unstructured correlation or a simple autoregressive correlation. While this model can be fit using either SAS (via PROC MIXED) or R (via the nlme package), it can also probably be fit using SPSS and perhaps using STATA as well.

Any linear model analysis will require diagnostics. At a minimum this will require graphical evaluation of the residuals and random effects to evaluate the qualities of homoscedasticity, independence, and symmetry. Normality can be evaluated using graphical evaluation of a normal quantile-quantile plot or by the use of a Shapiro-Wilk test.

The authors should plot the raw data from the trial either as a single spaghetti plot from pre to post with sequence indicated in color (i.e., both sequences superimposed) or in a separate panel for each sequence (i.e., both sequences juxtaposed). This should be done for each endpoint. These can be included as supplementary figures.

For endpoints that do not allow a simple spaghetti plot, the authors can plot individual subject profiles. In fact, this is highly recommended for the authors' own use anyway. It allows not only identification of outlying observations, but also identification of outlying subjects or outlying time points.

Note that randomization in a crossover design is not to groups, but to sequences.

Note that "repeated measures mixed models" is not a clear description. Instead, the authors need to describe the fixed effects, the random effects and their G-side covariance, and the R-side correlation structure used. If these are different for different endpoints, then the endpoints should be grouped as needed, and the model structure then detailed for each set of endpoints.

Note that models are not "performed", but are instead "fit", "used to assess", or the like. The same goes for tests of association --- these are not "performed" on data.

Rather than describing the analysis as "Statistical test X was used for categorical data", please list the endpoints followed by their statistical assessment, viz. "The association of treatment with endpoints X, Y, and Z was assessed using Fisher's exact test", or similar language.

The authors note that reductions were shown on the "financial difficulties" scale. This really seems to underscore the lack of any interpretable effects. This is not a criticism of the lack of effects, as it is equally important to publish the results of studies that "fail" to show effects as it is to publish studies that show effects. But, it is hard to see how the treatment would relate to this particular endpoint.

In general, please do not summarize results as "(all P>0.21)" and the like. The p-value is not that important of a measure, and anyway summarizing it in this way is only very slightly more informative than not summarizing it, at best. The results are in the table for the reader to see.

The authors write "indicating that t-VNS could be capable to module the CVT in a well-defined sub-group". Leaving aside the syntax error, this statement is a wild over-reach by the authors, considering their results and the methodology used (evidently) to get to this statement.

The authors note that "several types of analgesics" were in use. Acknowledging that it is unethical to control this usage in this setting, this seems like a potentially highly confounding factor. The authors should attempt at to quantify or at least assess the impact of this factor.

Table 1 should be presented also by sequence. Alternatively, the by-sequence breakdown could be presented in a supplementary table.

Table 2 is really confusing. Why are the sample sizes different for the various analyses? Why are some analyses labelled "supplementary"? What is being averaged over in the various columns? Perhaps Table 2 should be broken up into different tables, where the endpoints are similarly structured (i.e., single measurements post treatment, post versus pre, or more extensive repeated measures within period). The authors may wish to look at industry tables for presenting results from crossover designs.

Table 2 statistical tests should be presented pursuant to a proper analysis taking into account the crossover design and potentially the within-period repeated measures.

Table 4 should be reformatted to industry standards. The current layout is very poor.

In addition to Figure 3 it would be useful to see the data broken out by sequence, to see if there is anything like a carryover effect. The authors may wish to consider the possibility that this figure represents regression to the mean, since patients were required to have long-lasting pain before enrollment.

*** References: ***

Senn, S. J. (2000). Crossover design. Marcel Dekker, New York.

6. PLOS authors have the option to publish the peer review history of their article (what does this mean?). If published, this will include your full peer review and any attached files.

Reviewer #1: **Yes: **Eugenijus Kaniusas

Reviewer #2: No

Reviewer #3: No

---

## [Author Response · Author response to Decision Letter 0]

7 Nov 2020

Response to reviewer #1 comments:

The reviewer wrote #1: “The study describes effects of the cervical transcutaneous vagus nerve stimulation in chronic pancreatitis patients. Irrespective of negative findings, this study design and results are of high value for the interested reader.”

Our reply: Thank you very much for your kind remarks and acknowledgement of our study.

The reviewer wrote #2: „Transcutaneous vagal neuromodulation“ is too general, leading potentially to confusion with the auricular transcutaneous vagal neuromodulation. I would recommend to integrate either “cervical” or “on the neck”

Our reply: We agree that it is more precise to include “cervical. The title has now been changed to: “The effects of two weeks cervical transcutaneous vagal neuromodulation in chronic pancreatitis patients: A randomised sham controlled clinical trial”. 

The reviewer wrote #3: “Please add to the methodology section “cervical” or “on the neck”. Currently the reader of the abstract has no clue where the stimulation was performed.”

Our reply: We have now specified where the stimulation was performed by adding “cervical” in the abstract and the method section. Abstract: “Patients were randomly assigned to receive a two-week period of cervical transcutaneous vagal nerve stimulation using the gammaCore device followed by a two-week sham stimulation, or vice versa.” 

Method section: “This study was an investigator-initiated, randomised, double-blinded, sham-controlled, crossover clinical trial assessing the effect of cervical nVNS for pain treatment in patients with CP.”

The reviewer wrote #4: The abbreviation t-VNS is a bit misleading since tVNS is usually used for the transcutaneous auricular stimulation, just in contrast to pVNS as used for the percutaneous auricular stimulation. The stimulation on the neck is usually abbreviated as nVNS. It is a suggestion to avoid confusion of the readership.

Our reply: We have now changed the abbreviation t-VNS to nVNS as suggested throughout the entire manuscript.

The reviewer wrote #5: Ad Methods: I think that 2 weeks stimulation is a relatively short period, especially in view of a long-lasting chronic ailment with the associate changes in plasticity. For instance, the auricular VNS is usually applied for 6 weeks in chronic ailments such as the chronic back pain (https://doi.org/10.1213/01.ane.0000107941.16173.f7). Even though it is shortly discussed in the limitations section, the reader would welcome a more detailed discussion on the effective treatment durations of VNS and cervical transcutaneous VNS (optionally, including also auricular VNS) in chronic ailments. Please discuss and justify this issue in detail.

Our reply: This is a valuable point, and we agree with the reviewer that two weeks stimulation may be a relatively short treatment period to improve chronic pain. However, previous prospective, double-blind, sham-controlled, randomized studies with transcutaneous VNS have demonstrated that cervical transcutaneous VNS (with gammaCore) improved pain relief (compared to sham treatment) at 30, 60 and 120 minutes in migraine patients [Tassorelli et al. Neurology. 2018; 91,4]. Therefore, we expected that two weeks could be enough to induce a long-term pain relieving effect in patients with chronic pancreatitis. The study conducted by Sator-Katzenschlager et al. [Sator-Katzenschlager et al. Anesth Analg. 2004; 98,5] was performed in chronic back pain (somatic pain and not visceral pain) which may require a different treatment duration than chronic visceral pain. Also, the study did use auricular electroacupuncture and not cervical transcutaneous vagal nerve stimulation. Finally, the treatment was performed for 6 weeks, however the electrical stimulation was only given once weekly, in order to assess both short term and long-term benefit of electrical acupuncture. In contrast, our study aimed to assess a short-term effect of transcutaneous vagal nerve stimulation. Therefore, caution must be applied when comparing to previous VNS studies. 

As requested, we have now specified the rationale for choosing two weeks treatment in the method section: “Two weeks stimulation was chosen, since previous prospective, double-blind, sham-controlled, randomized studies with transcutaneous VNS have demonstrated that stimulations with gammaCore induced a pain reliving effect at 30, 60 and 120 minutes in migraine patients [13]. Therefore, we expected that two weeks would be sufficient to induce a long-term pain-relieving effect.”

Furthermore, we have now included a discussion regarding the treatment durations and type of VNS: “The duration of treatment seems to differ depending on the type of disease, type of intervention (invasive VNS, auricular or cervical nVNS), and time frame (short-term vs. long-term effect) [3]. For example, in chronic pain patients, an auricular nVNS study applied the treatment for six weeks [34], while a cervical nVNS study applied the treatment for two weeks [12]. Given that CP is characterized by long-lasting ailment, future studies may include a longer treatment duration to recover the functional plasticity in neural circuits of pain processing. In addition, the duration of the wash-out 

period of two weeks was chosen since it has been shown to be sufficient to reset the effects of neuromodulation [10], but long-term effects have also been demonstrated in patients with epilepsy [35]. However, if we extended the study duration (treatment and wash-out periods), a high risk of dropout would potentially occur, and study completion would become more difficult.”

The reviewer wrote #6: Please amend whether state-of-the-art individual treatment of patients was optimized before the treatment, interrupted or continued during the tVNS study. Address concomitant medication/treatment of patients which is very important since improvements were registered with respect to the baseline.

Our reply: We have now specified that all patients continued their usual medication during the entire study, but they were allowed to take additional medication in case of severe pain attacks. The method section is updated: “All patients were asked to maintain their usual pain medication and dosage during the entire study, and they were only allowed to take extra pain medication in case of severe pain attacks. Any changes needed in pain medication during the study period was noted by the patient in the pain diary. Patients were allowed to continue the following analgesics during the study period: paracetamol, nonsteroidal anti-inflammatory drug (NSAID), opioids, tricyclic antidepressant (TCA), serotonin-norepinephrine reuptake inhibitor (SNRI) and gabapentoids.”

Patients served as their own controls due to the cross-over design and they were asked to continue their usual medication during the entire study period. The sample size of the current study did not allow for a detailed subgroup analysis of the impact of concomitant pain medication but we agree that it would be relevant to investigate the impact of concomitant medication in future studies. Thus, we added the following in the discussion: “Furthermore, the impact of concomitant medication, the duration of CP or other disease characteristics (including stress, anxiety, and depression) could also be relevant to investigate. In future studies, a larger sample size is required in order to investigate sub-groups of patients, who possible could have an effect of nVNS.”

The reviewer wrote #7: Ad Method: Quantitative description of CVT is missing, even though a reference is given. Please describe it in a few sentences so that the reader has a clear idea about it.

Our reply: We have now specified the methods for acquisition of cardiac derived parameters and a description of the method used for calculation of CVT in the method section: “Next, ECG electrodes (Ambu Blue Sensor P, Denmark) were placed in the left and right subclavicular areas and cardiac apex and then attached to a non-invasive cardiac monitor. A biosignal acquisition system (ExtremeBiometrics, London, UK) was used to assess cardiac derived parameters from five minutes recordings.” “CVT was calculated beat-to-beat based on detecting positive phase shifts in the R-R interval and used as a putative measurement of heart rate variability [18]. Thus, low CVT values reflect low heart rate variability, whereas high values reflect high heart rate variability.”

The reviewer wrote #8: Ad Method: More information is needed on the wash-out period, how long, 1 week? With uninterrupted concomitant treatment.

Our reply: We thank the reviewer for this comment and fully agree that this information should be included. A two-week wash-out period was used in the current study. This is now specified in the method section: “The treatment periods were separated by a two-week wash-out period. The patients continued their usual analgesic medication during the wash-out period. The two week wash-out period has been used in previous studies of neuromodulation and shown to be sufficient to reset the effects of neuromodulation [10].”

The reviewer wrote #9: Ad Discussion: p.15 Discussed comparison/justification of cervical tVNS results with respect to invasive VNS results is limited due to the basically different technologies applied. Therefore, please address potential validation of cervical tVNS effects with respect to invasive VNS effects to substantiate your conclusions here. 

Our reply: We agree that cervical tVNS and invasive VNS are two different technologies, therefore caution must be applied when comparing those technologies. Hence, we have now addressed a potential validation of cervical tVNS effect with respect to invasive VNS in the discussion section: “Recently, a functional MRI study showed that cervical vagal afferents can be accessed non-invasively via transcutaneous electrical stimulation of the neck, which overlies the course of the vagus nerve [29]. Also, the brain activation elicited by nVNS was analogous to that elicited by invasive VNS, providing validation of cervical nVNS effects [27].”

 

Response to reviewer #2 comments:

The reviewer wrote #1: Abstract: specify that it was a cervical transcutaneous VNS. Specify the device (GammaCore). Our reply: We agree that this information should be presented in the abstract: “Patients were randomly assigned to receive a two-week period of cervical transcutaneous vagal nerve stimulation using the gammaCore device followed by a two-week sham stimulation, or vice versa.” Also, the title of the manuscript has been updated, specifying the use of cervical VNS. 

The reviewer wrote #2: The number of patients is rather small for me, as specified by the authors. 

Our reply: We agree with the reviewer that the sample size might be small, especially with regards to the secondary outcomes. We performed a sample size calculation based on the primary outcomes (VAS scores), revealing that 16 patients in a cross-sectional study should be sufficient to detect a clinical meaningful difference of 30% in the average daily pain score (primary endpoint). Thus, we consider that the number of patients was sufficient for the analysis of the primary outcome. Since no data exists on nVNS treatment in CP patients, the assumptions of the power calculation (standard deviation, etc.) was based on data from a study of patients with chronic pancreatitis patients, who received pregabalin treatment, which related to an improvement in clinical measures of the pain scores [Olesen et al. Gastroenterology. 2011;141,2]. However, we cannot exclude that our study was under-powered to detect an effect of VNS on the VAS score, since the true effect of VNS could be less than pregabalin. “The lack of pain improvement may reflect that a longer-lasting treatment is needed, but the study could also be under-powered to detect a potential effect.” We also acknowledge that it could be relevant to include more subjects in future studies in order to be able to investigate sub-groups of responders, impact of etiological factors, etc. We now added this to the discussion, now reading: “In future studies, a larger sample size is required order to investigate sub-groups of patients, who possible could have an effect of nVNS.” 

The reviewer wrote #3: - VNS stimulation is generally performed on the left VN while in the present study the authors stimulated both VN. Is-there any advantage to perform such a bilateral stimulation? Our reply: This is a very good point. To our best knowledge, there are no clear evidence or recommendations with regards to unilateral or bilateral transcutaneous VNS. Hitherto, previous transcutaneous VNS stimulation studies using the gammaCore device have stimulated bilaterally (on both left and right vagal nerve) in chronic pain patients [Tassorelli et al. Neurology. 2018; 91,e364-e373] and in immune-mediated inflammatory diseases [Tarn et al. Neuromodulation 2019;22,5]. One argument could be that both left and right vagal nerves need to be stimulated to ensure that the input from the transcutaneous VNS to the nucleus tractus sollitarius in the brainstem is sufficient in order to alter the afferent input to other regions of the central nervous system. It is uncertain whether unilateral stimulation with transcutaneous VNS is enough to modulate the brainstem and higher centers in the central nervous system. Future studies will indeed be needed to evaluate the optimal stimulation approach and investigate whether there is any difference between unilateral and bilateral stimulation.

In our study, in order to stimulate efficiently with transcutaneous VNS, and since we treated for only 2 weeks, we have chosen to replicate the same bilateral stimulation approach as the previous gammaCore VNS studies. Accordingly, we added the following in the limitations section of the discussion: “Firstly, the paradigm and parameters of nVNS may influence the effect of the treatment, including the treatment duration, number of daily stimulations, stimulation intensity, unilateral vs. bilateral stimulation, etc.”

The reviewer wrote #4: The duration of the study (2 weeks of stimulation) is most likely too short since VNS is a slow acting therapy as demonstrated for epilepsy. Indeed, VNS induces modifications of neuronal plasticity, which are progressive. CP is not comparable to “headache” which is FDA approved with the device GammaCore. As stated by the authors CP induces central modifications which can induce a central sensitivity syndrome. In addition, most of these patients use opioid compounds which are also able to induce central perturbation of pain. So, it is simplistic to think that a two-week VNS will be able to improve such patients. In addition, if the effect of t-VNS is delayed over time, when switching to sham stimulation, it could induce a bias of the results of sham stimulation. Our reply: This is a valuable point. We already elaborated more on the treatment duration in response to reviewer 1, question #5, and the discussion is now reading: “The duration of treatment seems to differ depending on the type of disease, type of intervention (invasive VNS, auricular or cervical nVNS), and time frame (short-term vs. long-term effect) [3]. For example, in chronic pain patients, an auricular nVNS study applied the treatment for six weeks [34], while a cervical nVNS study applied the treatment for two weeks [12]. Given that CP is characterized by long-lasting ailment, future studies should possibly include a longer treatment duration to recover the functional plasticity in neural circuits of pain processing.” 

We agree with the reviewer that a carry-over is a concern due to the cross-over design of our study. However, no carry-over effect was detected in the primary clinical endpoints in the revised statistical analysis (according to the reply to reviewer 3). Hence, it does not seem to be a significant bias of our study. To date, no other studies have conducted a cross-over study with the gammaCore device. Therefore, estimating the wash-out period was challenging. We agree that several VNS studies have shown long-term effect with VNS in patients with epilepsy. VNS treatment in epilepsy patients have been conducted with both invasive and non-invasive VNS, therefore it is important to differentiate between invasive VNS treatment and non-invasive transcutaneous VNS treatment. Most of the long-term VNS studies have been conducted with invasive VNS. The following is now added in the method section: “The treatment periods were separated by a two-week wash-out period. The patients continued their usual analgesic medication during the wash-out period. The two week wash-out period has been used in previous studies of neuromodulation and shown to be sufficient to reset the effects of neuromodulation [10].” The following was added in the discussion: “In addition, the duration of the wash-out period of two weeks was chosen since it has been shown to be sufficient to reset the effects of neuromodulation [10], but long-term effects have also been demonstrated in patients with epilepsy [35].”Finally, in the limitation section we addressed the risk of increasing the dropout rate if the study duration (treatment duration and wash-out period combined) was longer and the recruitment would become more difficult: “However, if we extended the study duration (treatment and wash-out periods), a high risk of dropout would potentially occur, and study completion would become more difficult.” 

The reviewer wrote #5: In addition, the sham stimulation is rather questionable since it induces a vibration which, as noted by the authors, is able to have an effect on the auricular branch of the VN. 

Our reply: We agree. As already mentioned in the limitation section, the sham device may provide some vibration and tactile sensation that potentially can stimulate the vagal nerve and induce an effect similar to the active treatment. This could also explain why there is no differences between the active and sham treatment, but a positive effect when comparing both the treatments to their respective baselines. The limitation section now reads: “Secondly, even though the sham device did not deliver electrical stimulation, it may not be physiologically inert as it provides some vibration and tactile sensation that potentially can stimulate the vagal nerve and induce an effect similar to the active treatment. This could also explain why there was no differences between the active and sham treatment, but a positive effect when comparing both the active and sham treatments to their respective baselines.”

The reviewer wrote #6: The authors did not discuss the parameters of VNS.

Our reply: There are many options for vagal nerve stimulation, such as frequency, amplitude, shape and duration of the electrical pulse. We used the fixed parameters of VNS related to the gammaCore device as specified in the method section. We agree that the parameters of VNS may influence the effect of treatment, as well as the treatment duration, number of daily stimulations, the stimulation intensity, unilateral vs. bilateral stimulation, etc. As this type of treatment is relatively new, the “optimal” study design and parameter settings are still not determined and need to be investigated further. We now added the following to the limitations section: “Firstly, the paradigm and parameters of nVNS may influence the effect of treatment, including the treatment duration, number of daily stimulations, stimulation intensity, unilateral vs. bilateral stimulation, etc.”

The reviewer wrote #7: The duration of CP is 8.1 ± 7.6 years, so very variable. Thus, the central integration of pain and central sensitization is supposed to be patient-dependent and different between patients. Our reply: This is correct. We are aware that the pain pattern and pain duration were not the same for all patients. Therefore, the degree of central sensitization, and also the potential benefit of VNS, could be patient-dependent as the reviewer pointed out. Indeed, future studies should take the degree of central sensitization into account (by sensory profiling, QST, brain MRI, etc.) to identify characteristics predicting sub-groups of patients that could specifically benefit from nVNS. We now added the following in the limitation section in the discussion: “Furthermore, the impact of concomitant medication, the duration of CP or other disease characteristics (including stress, anxiety, and depression) could also be relevant to investigate. In future studies, a larger sample size is required in order to investigate sub-groups of patients, who possible could have an effect of nVNS.” 

The reviewer wrote #8: The analgesic treatments may interfere with VNS (see above) 

Our reply: We agree. The use of analgesic drugs during the study could have an impact of degree of pain relief induced by nVNS. As also replied to reviewer 1, question #6, we have now specified that all patients continued their usual medication during the entire study, but they were allowed to take additional medication in case of severe pain attacks. The method section is updated: “All patients were asked to maintain their usual pain medication and dosage during the entire study, and they were only allowed to take extra pain medication in case of severe pain attacks. Any changes needed in pain medication during the study period was noted by the patient in the pain diary. Patients were allowed to continue the following analgesics during the study period: paracetamol, nonsteroidal anti-inflammatory drug (NSAID), opioids, tricyclic antidepressant (TCA), serotonin-norepinephrine reuptake inhibitor (SNRI) and gabapentoids.”

Patients served as their own controls due to the cross-over design and they were asked to continue their usual medication during the entire study as discussed above. This likely limits the effect of concomitant pain medication on the study results. However, we agree that it would be relevant to investigate the impact of concomitant medication in future studies powered for subgroup analyses. Thus, we added the following in the discussion: “Furthermore, the impact of concomitant medication, the duration of CP or other disease characteristics (including stress, anxiety, and depression) could also be relevant to investigate. In future studies, a larger sample size is required in order to investigate sub-groups of patients, who possible could have an effect of nVNS.”

The reviewer wrote #9: Pain is multidimensional and rather subjective, so the use of a VAS is questionable, although I agree that VAS is often used in such conditions. It seems to me that there are no questionnaires evaluating stress (perceived), depression, anxiety, coping strategies in this trial. 

Our reply: We agree, and we think that this could be very relevant and interesting to investigate in future studies. In this study we focused on pain intensity and quality of life but we acknowledge that additional outcome measures focused on other pain dimensions would have been relevant. However, the number of outcome measures need to be balanced against feasibility concerns and by inclusion of numerous questionaries patients may be fatigued and thus provide less valid responses. Notwithstanding this limitation, the QOL questionnaire includes several components reflecting cognitive, emotional and social functioning to some degree. Future studies could be optimized based on our knowledge from this present study and include some of the mentioned components. We added the following in the limitations section: “Furthermore, the impact of concomitant medication, the duration of CP or other disease characteristics (including stress, anxiety, and depression) could also be relevant to investigate.”

The reviewer wrote #10: Did the patients used or have used complementary medicines which are able to impact the pain factor? Our reply: We have now, as stated in the response to your question #8, specified that all patients continued their usual medication during the entire study, and were allowed to take additional pain medication only in case of severe pain attacks. This is now addressed in the manuscript according to the reply to your question #8 and reviewer 1 question #6. 

The reviewer wrote #11: A limit for me of non-invasive VNS by comparison to invasive one is the compliance. How the authors controlled that patients had regularly performed VNS (and bilaterally)? Our reply: We agree. The patients were instructed to register each stimulation intensity for the stimulation on both sides of the neck and as stated in the method section: “patients registered the intensity of the stimulation after each treatment in the pain diary”. Furthermore, as also stated in the methods section: “compliance was assessed by reading the remaining stimulations on the device display after each treatment period.” Thus, missing stimulations (non-compliance) could very likely be identified. In the end we have to rely on the patient’s willingness to provide the correct information. Information on compliance is provided in the result section, reading: “In the nVNS treatment, 98%±3% of all stimulations were applied compared with 94%±11% in the sham treatment (P=0.24). Furthermore, the average intensity of the stimulations was 31.8±6.9 units during nVNS treatment and 36.0±7.9 units for sham treatment (P=0.02).”

The reviewer wrote #12: Regarding vagal tone, did the authors include patients with only low vagal tone which should be more appropriate and is supposed to better beneficiate of VNS. It is not clear for me in the manuscript. Our reply: This is a very good point. We did not use CVT (assessment of vagal tone) as a screening tool for inclusion. An exclusion criterion was known peripheral neuropathy, but some patients could potentially have autonomic neuropathy which could make it more difficult to modulate the parasympathetic tone. A study by Liu et al. found that epilepsy patients having reduced heart rate viability compared to healthy controls, had significantly lower effect of VNS treatment (non-responders) [Liu et al. Epilepsia. 2017;48,6]. Thus, it seems likely that the potential effect of nVNS could be related to the degree of autonomic dysfunction. This could be relevant to investigate in future studies. We now clarified this in the discussion: “Despite one of the exclusion criteria was known peripheral neuropathy, some patients could have autonomic neuropathy. Hence, some patients could have vagal neuropathy, which can make it more difficult to modulate the parasympathetic tone and potentially mediating an anti-hyperalgesic effect [7]. A study by Liu et al. found that epilepsy patients having reduced heart rate viability compared to healthy controls, had significantly lower effect of VNS treatment (non-responders) [31]. Thus, the effect of nVNS could be related to the degree of autonomic dysfunction, and CVT may be useful as a screening tool in future studies.”

The reviewer wrote #13: The authors did not explore the anti-inflammatory effect of VNS which is well known in different chronic inflammatory conditions such as rheumatoid arthritis and Crohn’s disease. That would have been interesting and there is a relation between inflammation and pain. This point is only stated in the “Limitations” of the study. 

Our reply: We agree. Exploration of the anti-inflammatory effect of VNS could be very interesting and could add more valuable information about the nVNS treatment. According to our published protocol in BMJ open [Muthulingam et al. BMJ Open. 2019;9,7], we have collected blood samples to assess the anti-inflammatory effect of this nVNS treatment, and we are very excited to look further into these data. 

The reviewer wrote #14: Ongoing alcohol dependence was an exclusion criterion but not smoking? Nicotine is able to act on alpha7 nicotinic acetylcholine receptor and to have an anti-inflammatory effect. Our reply: We agree that smoking may have an impact on the anti-inflammatory effect. However, adding smoking as an exclusion criterion may lead to difficulties in recruiting CP patients in a 8-week clinical trial. In our study (see Table 1) almost 80% of patients were smokers and to enroll only non-smokers would have made it virtually impossible to recruit an adequate number of patients. We have included the following in the discussion: “Fourthly, even though the study was based on a cross-over design, where patients served as their own control, bias could be present due to the inclusion/exclusion criteria. For instance, ongoing alcohol dependence was an exclusion criteria whereas ongoing smoking was not, but nicotine could potentially impact the inflammatory-system [36].” 

 

Response to reviewer #3 comments:

Reviewer #3 (General remarks): The study follows a two period two treatment crossover design. However, the analysis which was performed is too simplistic and fails to account adequately for the actual trial design. 

The text itself is marred somewhat by a variety of minor syntactical, grammatical, and English usage errors. Since the authors did not provide page and line numbers as requested by PLOS ONE submission guidelines, no specific corrections will be offered.

Our reply: We thank the reviewer for these general remarks. We agree that our previous analyses did not adequately take the cross-over design into consideration in particularly regarding the analysis of carry-over effects. We have now consulted a senior statistician at our institution who agreed with the reviewer that a linear mixed effects model with fixed effects for period and treatment, and random effects for sequence and subject nested in sequence would be an appropriate statistical method. However, even thought our study was block randomized, the final allocation into the two sequences sham/VNS and VNS/sham was not balanced as most dropouts occurred in the latter sequence group. Consequently, a linear mixed effect model was not optimal for the structure of our data. As an alternative, the senior statistician advised us to use an analytic approach based on t-tests taking into account the cross-over design [Wellek et al. Dtsch Arztebl Ind. 2012;109,15]. This approach follows the guidelines recommended for analysis and reporting of crossover trials published in PLoS One [Li et al. PLoS One. 2015;10,8]. Overall, the results and conclusions obtained from the revised analysis are the same (i.e. a largely negative study). However, we are very grateful for the comments on the statistical analysis, which lead us to work more further with our data. 

Since we have not utilized a mixed linear model for analysis, the specific questions related to this model (see below), are not applicable for our current analysis, and consequently not addressed in detail below.

We have revised the method and result sections and the tables in the manuscript according to the new analyses. We have addressed the remaining questions below and finally, we have now revised the entire manuscript thoroughly according to syntactical, grammatical and English usage errors. 

The reviewer wrote #1: 

The usual modern analysis for a crossover design involves using a linear mixed effects model with fixed effects for period and treatment, and random effects for sequence and subject nested in sequence. The analysis of variance results should be presented, as well as effect estimates. Consult Senn (2000) for the details.

Our reply: Please refer to the comments provided for the general remarks above. We have revised the method and result sections, and the tables, according to our new statistical analyses. Overall, the results and conclusions are the same.

The reviewer wrote #2: 

For endpoints that the authors wish to model in finer detail, such as those involving measurements taken over time during each period, the authors may consider either summarizing the endpoints or elaborating the linear mixed effects model with within-period measurements. Summarizing is a less satisfactory method, because inference is then only to the summary measures rather than to the actual assessments of interest. The dependence due to repeated measurements over time should be accounted using an appropriate R-side correlation structure with either an unstructured correlation or a simple autoregressive correlation. While this model can be fit using either SAS (via PROC MIXED) or R (via the nlme package), it can also probably be fit using SPSS and perhaps using STATA as well. 

Our reply: Please, see the reply to the general remarks. We have now provided estimates with 95% confidence intervals for the key endpoints in the abstract and result section.

The reviewer wrote #3: Any linear model analysis will require diagnostics. At a minimum this will require graphical evaluation of the residuals and random effects to evaluate the qualities of homoscedasticity, independence, and symmetry. Normality can be evaluated using graphical evaluation of a normal quantile-quantile plot or by the use of a Shapiro-Wilk test.

Our reply: We have checked the normality using Shapiro-Wilk test. All the variables are normally distributed. The following is added in the Method section: “The continues variables were checked for normality using Shapiro-Wilk test.”

The reviewer wrote #4: The authors should plot the raw data from the trial either as a single spaghetti plot from pre to post with sequence indicated in color (i.e., both sequences superimposed) or in a separate panel for each sequence (i.e., both sequences juxtaposed). This should be done for each endpoint. These can be included as supplementary figures. For endpoints that do not allow a simple spaghetti plot, the authors can plot individual subject profiles. In fact, this is highly recommended for the authors' own use anyway. It allows not only identification of outlying observations, but also identification of outlying subjects or outlying time points. 

Our reply: We agree, that single spaghetti plots can provide important information and we now provided these plots for the primary endpoints in the supplementary materials (S3, Figure S1). The result section is reading: “No significant differences were observed in the maximal or average pain score between two weeks nVNS treatment and sham treatment (Figure 3 and supplementary material S3 for individual plots).” 

The reviewer wrote #5: Note that randomization in a crossover design is not to groups, but to sequences.

Our reply: We thank the reviewer for noticing the error. We have now replaced “groups” with “sequences” in the manuscript. 

The reviewer wrote #6: Note that "repeated measures mixed models" is not a clear description. Instead, the authors need to describe the fixed effects, the random effects and their G-side covariance, and the R-side correlation structure used. If these are different for different endpoints, then the endpoints should be grouped as needed, and the model structure then detailed for each set of endpoints.

Our reply: Please, see the reply to the general remarks. The statistical description has now been updated according to the new method used for statistical analysis. 

The reviewer wrote #7: Note that models are not "performed", but are instead "fit", "used to assess", or the like. The same goes for tests of association --- these are not "performed" on data. 

Our reply: Thank you for pointing this out. We have now revised the manuscript according to the proposed terminology. 

The reviewer wrote #8: Rather than describing the analysis as "Statistical test X was used for categorical data", please list the endpoints followed by their statistical assessment, viz. "The association of treatment with endpoints X, Y, and Z was assessed using Fisher's exact test", or similar language.

Our reply: We have now updated the statistical section and specified the following: “Categorical data (PGIC and adverse events) were analyzed using a McNemar test.”

The reviewer wrote #9: The authors note that reductions were shown on the "financial difficulties" scale. This really seems to underscore the lack of any interpretable effects. This is not a criticism of the lack of effects, as it is equally important to publish the results of studies that "fail" to show effects as it is to publish studies that show effects. But, it is hard to see how the treatment would relate to this particular endpoint.

Our reply: We agree that it is hard to see how this short-term treatment would relate to “financial difficulties”. According to the updated manuscript, the revised statistical analysis showed no significant differences for “financial difficulties”. 

The reviewer wrote #10: In general, please do not summarize results as "(all P>0.21)" and the like. The p-value is not that important of a measure, and anyway summarizing it in this way is only very slightly more informative than not summarizing it, at best. The results are in the table for the reader to see.

Our reply: We agree, the summarized results can now be seen in the tables and only few specific p-values are provided in the main text along with corresponding effect estimates. 

The reviewer wrote #11: The authors write "indicating that t-VNS could be capable to module the CVT in a well-defined sub-group". Leaving aside the syntax error, this statement is a wild over-reach by the authors, considering their results and the methodology used (evidently) to get to this statement.

Our reply: Thank you for this comment. We agree that the statement is overreached and with our new analysis approach, only heart rate was significantly different between treatments. We have now rephrased the sentence in the Discussion section: “Despite unsuccessful enhancement of mean CVT in our study, nVNS could potentially be able to module the parasympathetic tone in a sub-group of patients.”

The reviewer wrote #12: The authors note that "several types of analgesics" were in use. Acknowledging that it is unethical to control this usage in this setting, this seems like a potentially highly confounding factor. The authors should attempt at to quantify or at least assess the impact of this factor.

Our reply: We agree, that the use of medication may influence the effect of nVNS. As the study was designed as a randomized cross-over study, and as patients were asked to continue their usual medication during the entire study (they were allowed to take additional medication in case of severe pain attacks), we intended to minimize the influence of medication between sessions. However, we agree that the potential interaction between different types of analgesics and the effect of nVNS should be investigated further in future studies, which of course need to be designed and powered for that purpose. We included the following statement in the limitations section: “Finally, various confounding factors are present in this patient cohort. For instance, the patients used several types of analgesics which may influence the effect of nVNS and as already discussed it may be difficult to modulate the parasympathetic tone for some patients. Furthermore, the impact of concomitant medication, the duration of CP or other disease characteristics (including stress, anxiety, and depression) could also be relevant to investigate. In future studies, a larger sample size is required in order to investigate sub-groups of patients, who possible could have an effect of nVNS.” 

The reviewer wrote #13: Table 1 should be presented also by sequence. Alternatively, the by-sequence breakdown could be presented in a supplementary table.

Our reply: We agree, and this information is now implemented in the revised tables (Table 1 and Table S1 in supplementary material S2). 

The reviewer wrote #14: Table 2 is really confusing. Why are the sample sizes different for the various analyses? Why are some analyses labelled "supplementary"? What is being averaged over in the various columns? Perhaps Table 2 should be broken up into different tables, where the endpoints are similarly structured (i.e., single measurements post treatment, post versus pre, or more extensive repeated measures within period). The authors may wish to look at industry tables for presenting results from crossover designs. Table 2 statistical tests should be presented pursuant to a proper analysis taking into account the crossover design and potentially the within-period repeated measures. 

Our reply: Thank you for this comment. We have now simplified Table 2 and updated according to the revised statistical analyses. 

The reviewer wrote #15: Table 4 should be reformatted to industry standards. The current layout is very poor.

Our reply: Thank you for this observation. We now improved the layout of Table 4.

The reviewer wrote #16: In addition to Figure 3 it would be useful to see the data broken out by sequence, to see if there is anything like a carryover effect. The authors may wish to consider the possibility that this figure represents regression to the mean, since patients were required to have long-lasting pain before enrollment.

Our reply: This is a great suggestion. We have now provided a new figure according to the reviewer’s suggestions, see Figure 3.

---

## [Decision Letter · Decision Letter 1]

2 Feb 2021

PONE-D-20-13163R1

The effects of two weeks cervical transcutaneous vagal neuromodulation in chronic pancreatitis patients: a randomised sham controlled clinical trial

PLOS ONE

Dear Dr. Frøkjær,

Thank you for submitting your manuscript to PLOS ONE. After careful consideration, we feel that it has merit but does not fully meet PLOS ONE’s publication criteria as it currently stands. Therefore, we invite you to submit a revised version of the manuscript that addresses the points raised during the review process.

Three experts in the field have reviewed your manuscript and have provided positive feedback.

The revisions that need to be made relate to your smaller sample size.

You state that your team found no difference between the groups in this study, however as your sample size is small, it is possible that your test is underpowered. You do state this limitation briefly in your manuscript. However it would be more correct to state throughout your manuscript that you found no evidence of a difference between groups (as opposed to finding evidence of no difference between groups). As such, please re-word some of your statements made in your manuscript as follows:

Lines 68-69:

Current statement:

"In this sham-controlled crossover study, two weeks transcutaneous vagal nerve stimulation did not induce pain relief in patients with chronic pancreatitis."

Suggested statement:

"In this sham-controlled crossover study, we found no evidence that two weeks transcutaneous vagal nerve stimulation induces pain relief in patients with chronic pancreatitis."

Lines 388-390:

Current statement:

"Our sham-controlled crossover study provided evidence that adjuvant treatment with two weeks nVNS did not induce pain relief as compared to sham treatment in patients with painful chronic pancreatitis"

Suggested statement:

"Our sham-controlled crossover study provided no evidence that adjuvant treatment with two weeks nVNS induces pain relief as compared to sham treatment in patients with painful chronic pancreatitis"

In addition, please add one to two sentences about your sample size limitation to the limitations section of your discussion.

Finally, we request that you revise the phrase "the effects of" in your manuscript title.

We look forward to receiving your revised manuscript.

Kind regards,

Susan Hepp, PhD

Academic Editor

PLOS ONE

Reviewers' comments:

Reviewer's Responses to Questions

**Comments to the Author**

1. If the authors have adequately addressed your comments raised in a previous round of review and you feel that this manuscript is now acceptable for publication, you may indicate that here to bypass the “Comments to the Author” section, enter your conflict of interest statement in the “Confidential to Editor” section, and submit your "Accept" recommendation.

Reviewer #1: All comments have been addressed

Reviewer #2: All comments have been addressed

Reviewer #3: All comments have been addressed

2. Is the manuscript technically sound, and do the data support the conclusions?

Reviewer #1: Yes

Reviewer #2: (No Response)

Reviewer #3: (No Response)

3. Has the statistical analysis been performed appropriately and rigorously? 

Reviewer #1: Yes

Reviewer #2: (No Response)

Reviewer #3: (No Response)

4. Have the authors made all data underlying the findings in their manuscript fully available?

Reviewer #1: Yes

Reviewer #2: (No Response)

Reviewer #3: (No Response)

5. Is the manuscript presented in an intelligible fashion and written in standard English?

Reviewer #1: Yes

Reviewer #2: (No Response)

Reviewer #3: (No Response)

6. Review Comments to the Author

Reviewer #1: (No Response)

Reviewer #2: (No Response)

Reviewer #3: (No Response)

7. PLOS authors have the option to publish the peer review history of their article (what does this mean?). If published, this will include your full peer review and any attached files.

Reviewer #1: **Yes: **Eugenijus Kaniusas

Reviewer #2: No

Reviewer #3: No

---

## [Author Response · Author response to Decision Letter 1]

3 Feb 2021

Response to comments:

The editor wrote #1: “The revisions that need to be made relate to your smaller sample size.

You state that your team found no difference between the groups in this study, however as your sample size is small, it is possible that your test is underpowered. You do state this limitation briefly in your manuscript. However it would be more correct to state throughout your manuscript that you found no evidence of a difference between groups (as opposed to finding evidence of no difference between groups). As such, please re-word some of your statements made in your manuscript as follows:

Lines 68-69: Current statement: "In this sham-controlled crossover study, two weeks transcutaneous vagal nerve stimulation did not induce pain relief in patients with chronic pancreatitis."

Suggested statement: "In this sham-controlled crossover study, we found no evidence that two weeks transcutaneous vagal nerve stimulation induces pain relief in patients with chronic pancreatitis."

Lines 388-390: Current statement: "Our sham-controlled crossover study provided evidence that adjuvant treatment with two weeks nVNS did not induce pain relief as compared to sham treatment in patients with painful chronic pancreatitis"

Suggested statement: "Our sham-controlled crossover study provided no evidence that adjuvant treatment with two weeks nVNS induces pain relief as compared to sham treatment in patients with painful chronic pancreatitis"”

Our reply: Thanks for this helpful comment. We agree, due to the smaller sample size, that it is more correct to conclude that we found no evidence of difference between groups/treatments. We now reviewed the entire manuscript to adopt this. We agree that this concerns the two concluding statements (lines 68-69 and 388-390), which have now been corrected as suggested. In the beginning of the Conclusion section, we also adjusted lines 280-281 now reading “Overall, we found no evidence of significant difference in pain relief between the two treatments” and lines 288-290 reading “In contrast to our hypothesis, we found no evidence that CP patients had a significant pain reduction comparing the nVNS and sham treatments”.

The editor wrote #2: “In addition, please add one to two sentences about your sample size limitation to the limitations section of your discussion.”

Our reply: We now added the sample size discussion in the beginning of the limitations section, now reading: “First, even though a sample size calculation was performed prior to this study, such calculations in clinical studies with experimental interventions are often based on some uncertain assumptions. Hence, we could have a too small sample with a risk of false negative results regarding our primary outcomes”.

The editor wrote #3: “Finally, we request that you revise the phrase "the effects of" in your manuscript title.

Our reply: We agree. The title is now rephrased avoiding the “effect of” statement. It now simply reads: “Cervical transcutaneous vagal neuromodulation in chronic pancreatitis patients with chronic pain: a randomised sham controlled clinical trial”.

---

## [Editor Report · Decision Letter 2]

11 Feb 2021

Cervical transcutaneous vagal neuromodulation in chronic pancreatitis patients with chronic pain: a randomised sham controlled clinical trial

PONE-D-20-13163R2

Dear Dr. Frøkjær,

We’re pleased to inform you that your manuscript has been judged scientifically suitable for publication and will be formally accepted for publication once it meets all outstanding technical requirements.

Kind regards,

Julia Robinson

Senior Editor

PLOS ONE
---

## [Editor Report · Acceptance letter]

17 Feb 2021

PONE-D-20-13163R2 

Cervical transcutaneous vagal neuromodulation in chronic pancreatitis patients with chronic pain: a randomised sham controlled clinical trial 

Dear Dr. Frøkjær:

I'm pleased to inform you that your manuscript has been deemed suitable for publication in PLOS ONE. Congratulations! Your manuscript is now with our production department. 

Kind regards, 

on behalf of

Julia Robinson 

Staff Editor

PLOS ONE